# Using Random Ordering in User Experience Testing to Predict Final User Satisfaction

**Kitti Koonsanit *, Daiki Hiruma, Vibol Yem and Nobuyuki Nishiuchi**

Department of Computer Science, Graduate School of Systems Design, Tokyo Metropolitan University, Tokyo 191-0065, Japan
* Correspondence: koonsanit-kitti@ed.tmu.ac.jp

**Abstract:** In user experience evaluation (UXE), it is generally accepted that the order in which users perform tasks when using a product is often random rather than fixed. UXE based on these so-called randomly ordered tasks is challenging. Although several articles have been published on UXE, none have proposed a technique to evaluate the significance of randomly ordered tasks. In this study, we propose a new approach to predict final user satisfaction based on UX related to randomly ordered tasks. We aimed to study the importance of task order in the UX. In the main experiment, 60 participants completed questionnaires about satisfaction while performing a series of tasks on a travel agency website. Among the machine learning models tested, we found that accounting for the order or sequence of actions actually performed by users in a support vector machine (SVM) algorithm with a polynomial kernel produced the most accurate predictions of final user satisfaction (97%). These findings indicate that some machine learning techniques can comprehend participants' randomly ordered UX data. Moreover, using random ordering, which accounts for the actual order of actions performed by users, can significantly impact the prediction of final user satisfaction.

**Keywords:** user experience; randomly ordered tasks; UX evaluation; satisfaction; prediction; machine learning



## 1. Introduction

Final user satisfaction based on user experience (UX) is critically important for product evaluation and users' decision-making about whether or not to continue to use or recommend a product or service to others [1,2]. Predicting final user satisfaction is thus key for many use cases in product evaluation.

Many product developers have attempted to evaluate final user satisfaction by gathering and analyzing users' historical behavior. One example of this can be found in the analysis of consumer behavior on websites. With each visitor possessing their own unique website usage habits, product developers have aimed to decode visitor's usage and apply the data to predict final user satisfaction. However, extrapolating final user satisfaction from visitors' website usage can be difficult. Among the variety of evaluation methods used to assess final user satisfaction, each with its own advantages and drawbacks, one of the most significant is user satisfaction analysis. Previous studies have reported that user satisfaction analysis helps product developers identify how final user satisfaction affects decisions related to a product or service [1]. Traditionally, product developers have relied on collecting information on user satisfaction from questionnaire surveys administered from the first to the final stage of usage [3]. In the case of a website, final user satisfaction prediction is also based on the volume of activity on a website in the form of page views and hits, and the order of visits using the website's cookie technology and user logs [4], as shown in Figure 1.

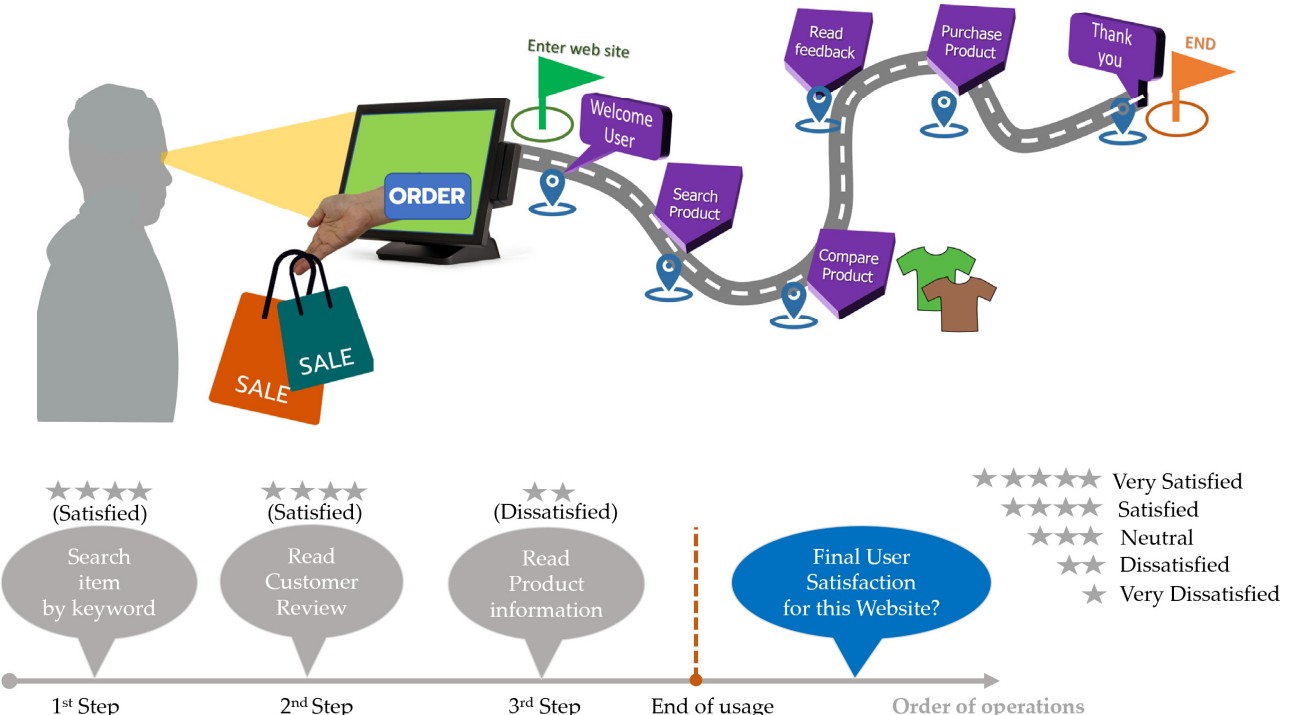

**Figure 1.** User satisfaction while users are on a shopping website.

User satisfaction analysis has been adopted to evaluate UX. Several studies have defined UX as any interaction a user has with a product or service [5,6]; for example, how the product looks, how its elements influence the user, how it makes them feel, and how they interact with it. Understanding momentary UX data can help determine final user satisfaction after usage. Recent studies have reported that UX is an important variable for building customer satisfaction post-purchase in the product industry [7,8]. All UX variables can affect a customer's satisfaction [9]. In UX evaluation (UXE), users' questionnaire responses are used to draw a customized UX Curve to represent changes in user satisfaction. After usage, the users are asked to complete a final questionnaire about their satisfaction with the product. Answers relating to final user satisfaction are often expressed on a scale of negative to positive values and are also often based on the order in which users interact with the product. Thus, user satisfaction assessments consider both the satisfaction score and the order of users' activities. Further, UXE is an important tool for improving the success of a product and forms a critical strategy by which to seek feedback from users with the goal of improving their final user satisfaction.

To have a great experience using a product or service, customers need to be able to easily access all the essential functions they need. In addition to being aesthetically pleasing, a product should also be simple and easy to navigate. Poor product design can discourage a user from spending time on the product. Given the vastness of the product industry, users who have a bad experience using one product can easily find what they need with another, and will not waste their time on products with bad UX. In contrast, a good UX will lead customers to recommend the products or services to other people. As a consequence, it is now clear to companies in many business sectors that it is impossible to design products and services without contributions from representative users.

However, a major problem with UXE is that it does not necessarily reflect real UX in terms of the order in which users perform tasks on a product. In UXE, a UX designer must design an order of tasks for the user to perform and related questionnaires beforehand. To measure UX during UXE, users are asked to follow a set of sequential tasks and complete the questionnaire about their feelings related to the product or service in a fixed order. In reality, however, it is impossible to predict what the user will and wants to do. Furthermore, it is generally accepted that different users will choose a different set of sequential tasks

to perform when using a product or service. This can clearly be seen when users visit e-commerce websites, where visitors can buy goods on any page on the site. While some visitors may navigate from the home page, others can do so from almost any other page on the site. Thus, in real life, the UX of a website involves randomly ordered tasks.

UXE based on the random ordering of tasks is challenging because of the difficulty related to classifying these data. Although a number of studies have conducted UXE of subjects, none have examined the sequence of actions in a UX that involves randomly ordered tasks. Further, to our knowledge, all available UXE approaches are based on a fixed sequential order of tasks, which means most UXE approaches are designed to examine actions performed in the same order, such as that reported in our previous research [1].

This study proposes a new approach using machine learning techniques to predict final user satisfaction based on UX related to randomly ordered tasks. We show that accounting for the sequence of actions may improve the prediction of final user satisfaction.

## 2. Literature Review

This section provides a brief overview of the relevant theoretical foundations of this study. We describe some background on UX evaluation methods, order effects in UX related to this work, and some details of general and relevant machine learning algorithms.

### 2.1. UX Evaluation Methods

As mentioned above, UX refers to all aspects of how people interact with a product or service. Many approaches have been proposed to evaluate UX. These methods are defined as being uniquely applicable to a specific period, namely before, during, or after usage. In 2011, Roto et al. published the User Experience White Paper, a document reporting Dagstuhl Seminar's categorization of UX from the viewpoint of the time axis [10]. The document underlines the importance of analyzing UX across time, and describes four types of UX—anticipated, momentary, episodic, and cumulative—each of which is defined based on usage time. Anticipated UX relates to the period before the first use; momentary UX relates to the period during usage and refers to any perceived change that occurs at the moment of interaction [11]; episodic UX relates to the period after usage; and cumulative UX relates to the entire period, from before the first use, during usage, and after usage. These four types of UX can affect final user satisfaction.

One way to conduct UXE is using a UX Curve, a method designed to facilitate the collection of past UX data [12,13]. The curve is used to help users retrospectively report how and why their experience with a product has changed over time and allows researchers to visualize interactions with the product from the customer's point of view. The curve is drawn on a horizontal axis showing the time and activities engaged in during usage and a vertical axis showing the satisfaction level (positive or negative) during usage. The satisfaction level can fluctuate significantly depending on the time and order of activities performed, as shown in Figure 2.

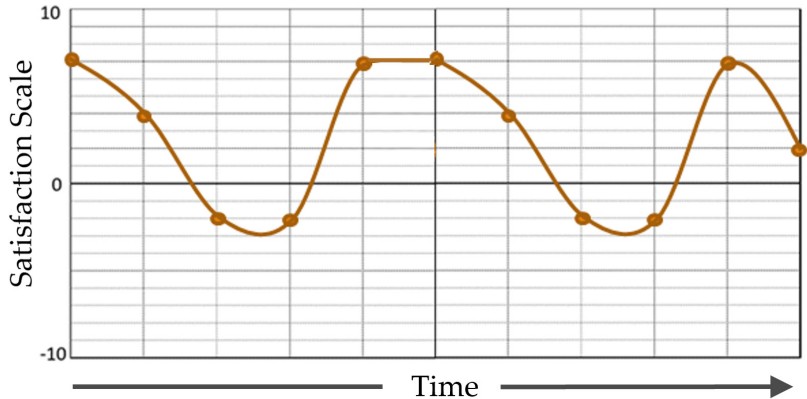

**Figure 2.** Example of a UX Curve.

Another UXE method and descendent of the UX Curve is the UX Graph [14,15]. Designed by Kurosu et al., the method involves plotting the user's satisfaction as an intensive measure of UX on one graph. Kurosu developed software to enable users to easily depict their degree of satisfaction on a time scale. The graph can be drawn after the user enters "episodes" describing their experience and satisfaction rating. User satisfaction is represented on the vertical axis, and final user satisfaction is defined as satisfaction after momentary UX, as shown in Figure 3.

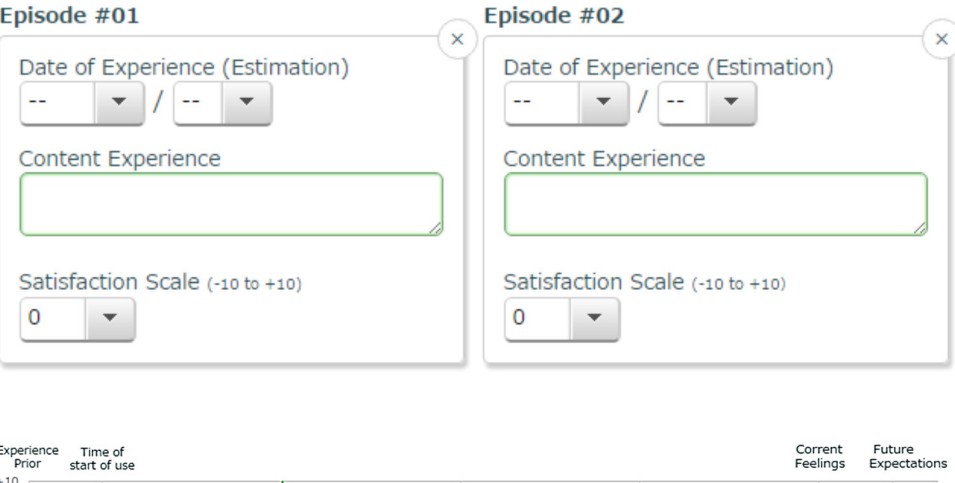

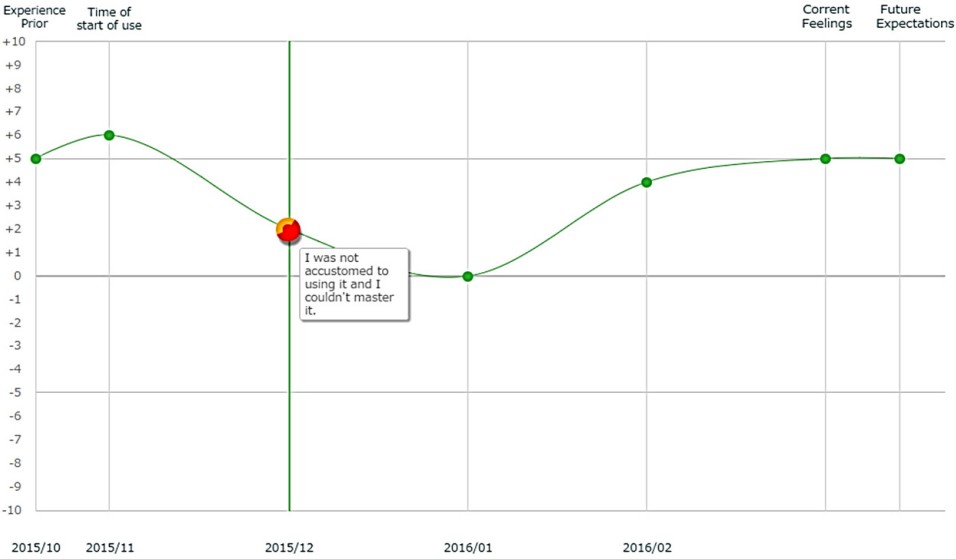

**Figure 3.** Users enter episodes comprised of their experience and satisfaction rating, and a UX graph showing these episodes is generated.

Several studies have measured UX to improve final user satisfaction [1,16–22]. In addition to the fact that momentary UX can affect episodic UX, various factors will interact in an intricate manner during the actual UX, and the final user satisfaction (episodic UX) will be determined from the accumulation of experiences. Sukamto et al. proposed an approach to enhance user satisfaction with a portal website by increasing the UX score in the User Experience Questionnaire (UEQ) [16]. The UEQ uses a questionnaire to measure users' subjective impression of their user experience of products. The UEQ is a semantic differential with 26 question items, with responses provided on a 7-point Likert scale from −3 (fully agree with the negative term) to +3 (fully agree with the positive term). Users will achieve a better score in the UEQ as they become more comfortable using the portal website. Based on the User-Centered Design method [16], the researchers assessed the initial and final usability score for a website using the UEQ. They showed that increasing the UX score through the UEQ in the development of a portal website could enhance customer satisfaction. In other related work, Pushparaja et al. examined the factors of

UX that influence users' satisfaction when using a digital library website [17]. Through a literature review, the team found that attractiveness, efficiency, dependability, stimulation, and novelty were key factors of satisfaction. Meanwhile, Mominzada et al. reported the consequences of UX and its role in a gamified e-commerce platform [18]. With the aim to identify the effects of gamification on user experience and the related consequences, the researchers used an online survey questionnaire as the main instrument for data collection. They showed that UX positively affects final user satisfaction in a gamified e-commerce platform. The experiment was statistically tested and validated through a quantitative research approach.

Due to differences in UX among users, both humans and computers have had difficulty classifying these data to develop or improve products and services. Furthermore, although responses to UX questionnaires can provide an abundance of information about a range of user feelings, the complexity of these data substantially increases the computational burden for experts. Machine learning techniques have recently been successfully used to make UX questionnaires easier to analyze and interpret. Many popular machine learning techniques have been used to analyze UX questionnaires, including support vector machine (SVM) classifiers, logistic regression, decision trees, and neural networks (NNs). Several studies have used machine learning to gauge final user satisfaction [1,19]. Research by Koonsanit et al. proposed an approach to predict final user satisfaction by combining momentary user experience data with machine learning techniques [1]. User satisfaction was measured while each user performed a fixed order of tasks on a product. The study reported that machine learning methods such as SVM can accurately predict final user satisfaction and contribute to developing better products and services by analyzing UX. SVM with a polynomial kernel had the highest cross-validation accuracy at 93%. The findings suggest that machine learning could be useful for analyzing momentary UX data to predict final user satisfaction. In other related work, Nwakanma et al. proposed a method for classifying the quality of UX and predicting customer sentiment to improve service delivery [19]. They collected UX data using Google Forms and developed an improved logistic regression classifier to test, train, and classify UX. The training accuracy of the proposed improved logistic regression was 97.67%, indicating the potential and capability of machine learning for analyzing a large database of sentiments or reviews and predicting customer sentiment. Machine learning approaches for UXE can thus be applied to industrial scenarios to evaluate users' perceptions of products and services.

In fact, Cong et al. proposed a machine learning-based iterative design approach to automate the prediction of user satisfaction called Smart PSS [22]. UX data in this study represented the subjective psychological feeling gained from the Smart PSS experience during use and were attained by calculating the user satisfaction score collected on a 5-point Likert scale, with 5 points indicating extreme satisfaction and 1 point indicating extreme dissatisfaction. Overall user satisfaction with Smart PSS was determined after users had completed 10 tasks. Data were collected from 20 test users in this study. After the experiment, the collected data were reorganized and processed using several techniques including data cleansing, data integration, feature selection, and data augmentation, before creating classification models. After building three models using SVM, decision trees, and an NN and comparing their performances, the researchers showed that the NN was the most accurate, with an overall accuracy of 90%. Meanwhile, models using SVM and decision trees had greater than 84% accuracy in the test set.

### 2.2. Order Effect in UX

In our literature review, we found that one major challenge of UXE is the order effect. Keiningham et al. reported that the order of activities and users' satisfaction with the most recent task in the UX significantly affect final user satisfaction [20]. They showed that the weights of initial satisfaction and transaction-specific satisfaction decay geometrically with time. This causes user satisfaction with the most recent task to receive more weight and priority when determining final satisfaction. Thus, more recent transaction-specific

satisfaction levels tend to have a greater influence on final satisfaction. In another interesting study, Min et al. proposed the importance of task order in a customer complaint system [21]. In general, it is often believed that employees should first apologize to customers before listening to their complaints. To test this belief, Min et al. created a digital library site and invited participants to perform article searches to evaluate the website. During their task, the participants encountered a service failure in the form of a long wait time caused by a slow response on the site. The researchers reported that participants with high expectations were more satisfied with a responsive apology (i.e., listen-and-then-apologize) than a preemptive apology (i.e., apologize-and-then-listen), with a mean satisfaction score of 4.64 and 3.85 ($p < 0.01$), respectively. Thus, a simple change in the sequential order of tasks such as apologizing and listening (listen-and-then-apologize) to complaints can significantly impact customer satisfaction and final user satisfaction.

The above-mentioned studies emphasize the challenges of UXE. The order of tasks in the UX and the user's memory of their satisfaction with the most recent task significantly affect their final satisfaction. The sequence and order of actions, including whether they are fixed or random, performed on a product or service are also important. Accounting for the order or sequence of actions could improve predictions of final user satisfaction. In the present study, we assumed that a change in the sequential order of tasks could significantly affect final user satisfaction.

### 2.3. Machine Learning

Determining final user satisfaction based on UX can be challenging because of the multifaceted nature of the UX. For example, UX is subjective, relating to an individual's feelings and past experiences. Most studies have used questionnaire systems to measure UX data in real time [3]. Based on the abundance of information that has been derived from randomly ordered tasks, UX data are not simply one-dimensional. Furthermore, performing multiple-evaluation comparisons can be difficult due to the variety of checklists used and the difficulty in quantifying expert opinions. UX also changes over time. Thus, it is often difficult and tedious for humans to interpret or classify the data to determine final user satisfaction. Humans are also prone to many types of bias. Further, as the accuracy of current analytical methods is often dependent on the level of expertise of the analyst, predictions of final user satisfaction can vary greatly among studies.

Several studies have been published on UX analysis. In 2022, Toshihisa et al. proposed that the peak-end rule was effective for evaluating UX [23]. They collected overall satisfaction ratings at each episode and used multiple regression analysis to verify the usefulness of the peak-end rule (explanatory variables: Maximum, minimum, and end score in a series of the episodes; objective variable: Overall satisfaction). Their analysis showed that the episode with the lowest satisfaction and the satisfaction at the end of the event were both correlated with overall satisfaction. However, only using peak-end data to evaluate UX may lead to the loss of valuable details typically available from the UX curve in predicting final user satisfaction.

In this study, we used adaptation in machine learning to solve the user satisfaction problem. Previous research [1] has shown the effectiveness of using machine learning on momentary user experience data to predict final user satisfaction. Despite the intrinsic challenges of individual algorithms, machine learning algorithms may produce fairer, more efficient, and bias-free outcomes than humans. Machine learning is advantageous as it facilitates analysis and understanding of the structural details of UX data [1].

Machine learning algorithms for UX classification represent a relatively novel approach to user satisfaction analysis. Some strategies used for classifying UX data include classical machine learning, statistical approaches, NNs, and pattern mining. The present study examined seven machine learning algorithms for the classification of UX data, namely random forest, K-nearest neighbors, SVM polynomial, SVM linear, SVM radial basis function (RBF), SVM Sigmoid, and AdaBoost.

According to previous research [1,22,24,25], machine learning algorithms such as SVM [26] can accurately predict final user satisfaction based on momentary UX data. A classical machine learning model works through a simple premise: It learns from the data with which it is fed. Collecting and feeding more data into a classical machine learning model leads to more training and more accurate predictions as it continually learns from the data. Previous studies have largely used supervised learning algorithms such as SVM for classification. As one of the most popular classical machine learning algorithms in use today [27], SVM is particularly effective in high-dimensional spaces. The effectiveness of an SVM depends upon the kernel function and parameters of the kernel.

*2.4. Sampling Techniques*

The number of data points is an important factor to consider when creating machine learning models. The issue of insufficient data has received considerable attention. Recent studies have examined the effect of sample size on machine learning algorithms. Although it can be difficult to determine the exact number of data points that a given algorithm requires, some studies have demonstrated that using small sample sizes for building classical machine learning models leads to better performance [28]. Other studies discuss the number of samples to use per class for small general datasets [29].

Insufficient data can lead to serious problems, such as an imbalanced distribution across classes [30]. Because many machine learning algorithms are designed to operate on the assumption that there are equal numbers of observations for each class, any imbalance can result in poor predictive performance, specifically for minority classes. To correct an imbalance in distribution, we performed a suite of data sampling techniques to generate alternative, synthetic data [31]. By generating samples of minority class data for the training dataset, the sampling method adjusts the class distribution of the dataset, which can then be used to create a new classification model with a machine learning algorithm. Several different sampling techniques have been reported for use with imbalanced datasets [31–35].

## 3. Materials and Methods

In our previous research [1], we designed an experiment with a fixed order of tasks in which participants were asked to complete all tasks, from the first to the final task, in sequential order. For example, in an online shopping evaluation, each user had to visit webpages A, B, C, and D in that order. However, this is not reflective of the real-life UX, where users are free to use the product however they want. Some users may visit the web pages in a different order to the specified A, C, B, and D, as shown in Figure 4 (right). We called this case randomly ordered, as opposed to fix ordered.

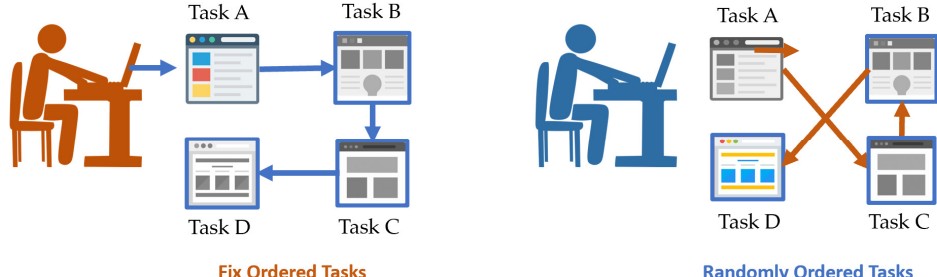

**Figure 4.** Comparison between fix ordered tasks and randomly ordered tasks performed when users use an online shopping website.

As shown in Figure 4, we used randomly ordered tasks. We had no control over when participants would visit any particular page. While some may start from the first page, others may start at a different page on the website, as shown in Figure 5.

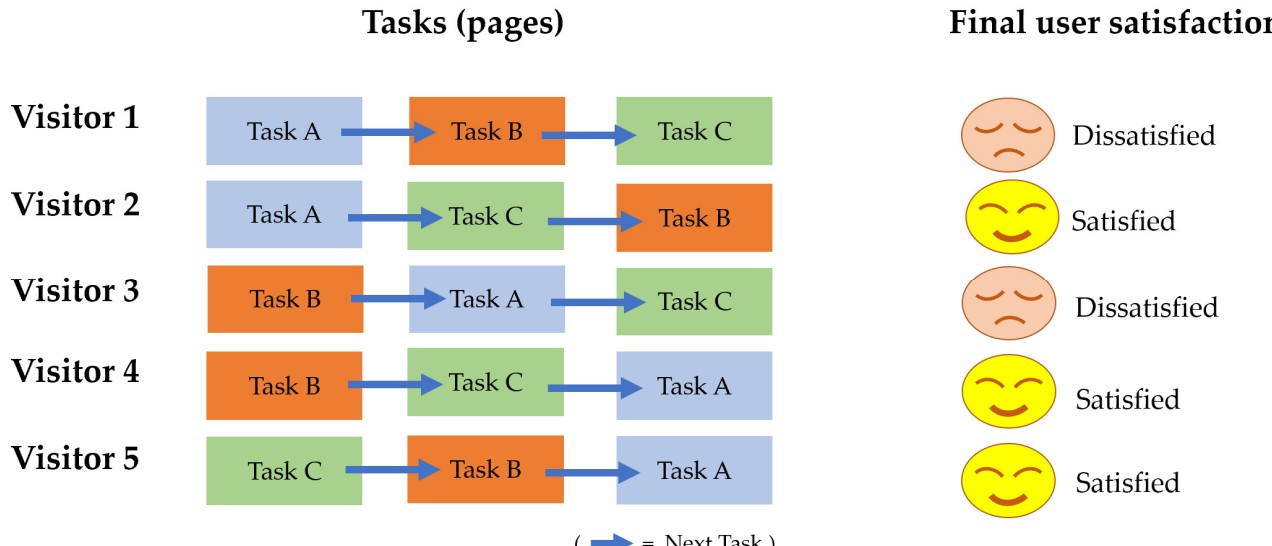

**Figure 5.** Example of randomly ordered tasks when users use an online shopping website.

*Proposed Framework*

In the UX approach, classification analytics-built models rely on random ordered UX data to predict user satisfaction levels. Our proposed framework aims to predict final user satisfaction guided by randomly ordered UX data to answer satisfaction-related questions. The evaluation process workflow architecture is shown in Figure 6.

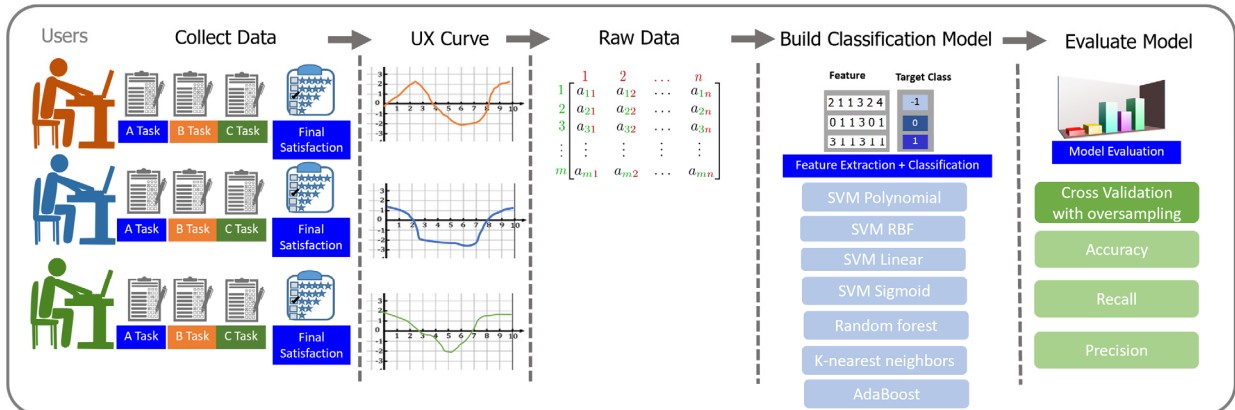

**Figure 6.** Workflow of our proposed evaluation process.

Our proposed framework was organized into three main steps. First, UX data were collected by gathering information and calculating scores in satisfaction survey questionnaires completed by users who performed randomly ordered tasks. Second, we built a machine learning framework to classify final user satisfaction into different classes. To confirm its effectiveness, the proposed framework was applied to an experiment using randomly ordered UX data from users' visits to a travel agency website. Randomly ordered UX data from the satisfaction survey questionnaire were used to represent changes in emotion. Finally, the classification model was evaluated using leave-one-out cross-validation and data-splitting techniques.

In our proposed framework, we wanted to compare the use of randomly ordered UX data between Dataset I, which did not account for the actual task order, and Dataset II, which did, under our specific conditions. Consequently, we needed to collect and generate our own original UX dataset. Details of the framework are explained in Section 4.2.

## 4. Experiments

In our previous study, we collected responses to a questionnaire about users' feelings related to a product or service while they performed fixed-order tasks [1]. In the present study, we used these data to simulate UX after shuffling the original order of tasks performed in the fixed-order experiment. The aim of these preliminary experiments was to compare the real-life results of the fixed-order task with the simulated results of the shuffled-order task.

Next, we performed the main experiment, in which users conducted all tasks in random order. The actual order or sequence of actions was recorded during usage.

The preliminary and main experiments are described in detail below.

### 4.1. Preliminary Experiments

We conducted preliminary experiments to determine the importance of task order in UX and how it can affect the final user satisfaction of customers while they are performing tasks on a website or product. To do this, we simulated UX after shuffling the order of tasks performed in our previous experiments [1]. The participants of the original experiments were university students who were asked to evaluate a travel agency website (Preliminary Experiment I) or Google Nest Mini smart speaker (Preliminary Experiment II) during usage by completing a satisfaction survey. Responses were used to draw a customized UX curve. Participants were also asked to complete a final satisfaction questionnaire about the product or service.

#### 4.1.1. Preliminary Experiment I: Travel Agency Website

In preliminary experiment I, we used data from a study in which participants were asked to evaluate a travel agency website [1]. The aim of the prior study was to predict final user satisfaction based on the satisfaction score given by users while using the website. Participants had to respond to seven satisfaction survey questions concerning the travel agency website. Before starting the task, we told them that their goal was to use the website to find a place they wanted to visit once in their lifetime. All participants appeared to perform the task attentively.

Fifty participants completed the six steps required to achieve the goal in a fixed order, allowing us to produce a customized UX curve depicting each one (steps 1–6). This procedure is often implemented in actual service or product usage to obtain the momentary UX, as shown in Figure 7. After completing the seventh step, the participants recorded their final satisfaction based on several experiences. The seventh step was conducted to obtain final user satisfaction as an indicator of episodic UX for this experiment only. The final satisfaction data obtained in step 7 were used as the target class variable for supervised learning. In this study, the number of classes was five.

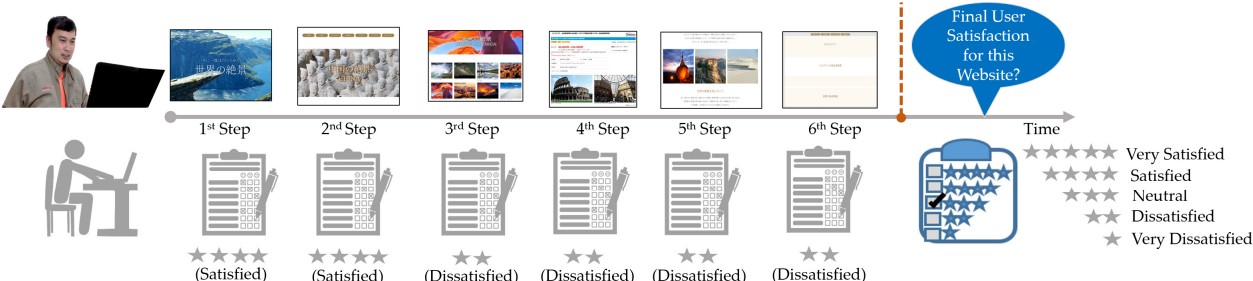

**Figure 7.** Evaluating user satisfaction while using a travel agency website.

As the aim of preliminary experiment I was to study the usefulness of sequential ordering for identifying factors predictive of final user satisfaction, we created two different datasets with which to build our machine learning models (Figure 8). Dataset W1 contained UX data that were simulated based on a shuffled order of tasks performed by participants in the real-life experiment reported in our previous study [1]. Meanwhile, Dataset W2

contained the original UX data that were obtained based on the actual order of the tasks performed on the travel agency website. This dataset had the same structure in terms of the fixed order of tasks as that reported in our previous research [1].

**Figure 8.** Example of the structure in each dataset.

In the evaluation step, we used polynomial SVM and leave-one-out cross-validation (LOOCV) to evaluate the performance of each dataset in the prediction of final user satisfaction. Table 1 shows the models' performance for the two datasets.

**Table 1.** Performance of prediction models of final user satisfaction for a travel agency website.

| Leave-One-Out Cross-Validation (LOOCV) | Dataset W1 (Shuffled Ordered of Tasks) | Dataset W2 (Actual Order of Tasks) |
|---|---|---|
| Cross validation accuracy without oversampling | 0.48 | 0.72 |
| Cross validation accuracy with oversampling (SMOTEN) | 0.58 | 0.90 |

The accuracy score ranges from 0.00 to 1.00; a higher value indicates higher accuracy.

### 4.1.2. Preliminary Experiment II: Google Nest Mini

In preliminary experiment II, we used data from a prior study in which participants evaluated the Google Nest Mini smart speaker [1]. The aim of the prior study was to predict final user satisfaction based on the satisfaction score given by users while they used the product. Participants had to respond to twelve satisfaction survey questions concerning the Google Nest Mini.

Twenty-five university students aged 21–24 years were recruited as participants. The task assumed that the participants had purchased the new smart speaker and removed it from its box. After removing the smart speaker from its box, the participants were required to set it up and start using it by performing 11 steps in a fixed order, as shown in Figure 9. The participants performed each step by referring to the enclosed instructions. At the end of each step, the participants recorded their satisfaction, which was then used to draw a customized UX curve. Their satisfaction with the product after completing each of the 11 steps was used as an indicator of momentary UX. The final user satisfaction obtained after completing all 11 steps was used to indicate episodic UX and as the target class variable for supervised learning. In this study, the number of classes was five.

In preliminary experiment II, we created two datasets (P1 and P2), as we did in preliminary experiment I, with which to build our machine learning models. Dataset P1 contained UX data that were simulated after shuffling the original order of tasks performed in the real-life experiment. Meanwhile, Dataset P2 contained the original UX data that were obtained based on the actual order of tasks performed on the Google Nest Mini in our previous research [1].

In the evaluation step, we used SVM polynomial and LOOCV to evaluate the performance of each dataset in the prediction of final user satisfaction. Table 2 shows the models' prediction performance for the two datasets.

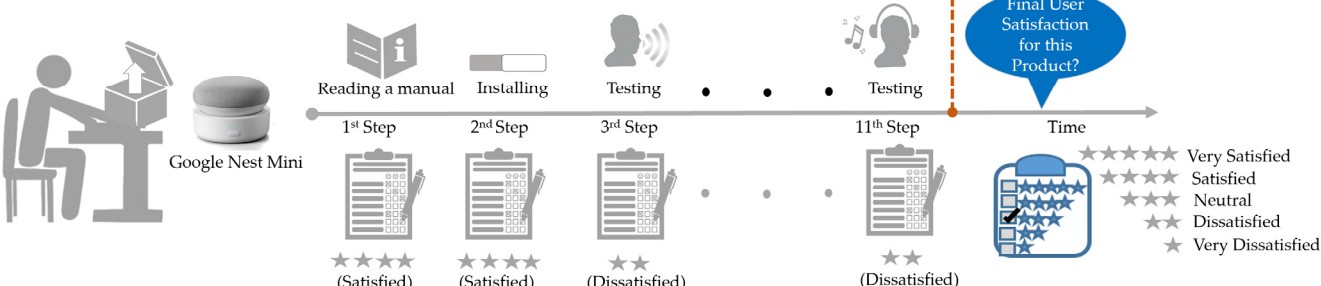

**Figure 9.** Evaluating user satisfaction while users set up the Google Nest Mini.

**Table 2.** Performance of prediction models of final user satisfaction for the Google Nest Mini smart speaker.

| Leave-One-Out Cross-Validation (LOOCV) | Dataset P1 (Shuffled Ordered of Tasks) | Dataset P2 (Actual Order of Tasks) |
|---|---|---|
| Cross validation accuracy without oversampling | 0.56 | 0.60 |
| Cross validation accuracy with oversampling (SMOTEN) | 0.64 | 0.76 |

The accuracy score ranged from 0.00 to 1.00; a higher value indicates higher accuracy.

As noted above, we wanted to understand the importance of task order while a user uses a product or service. Thus, after the evaluation step, we compared the two datasets. Our comparison demonstrated that the sequence or order of actions performed on a product or service, namely whether they were the actual fixed order of tasks or a shuffled order of the tasks, is important. From preliminary experiments I and II, we found that Datasets W2 and P2, which contained the original UX data obtained based on the actual order of tasks performed by participants, produced the most accurate predictions of final user satisfaction, at 0.90 and 0.76, respectively. Integration of the actual order of actions into a dataset can thus affect a model's prediction of final user satisfaction.

The results of preliminary studies I and II indicate that a change in the sequential order of tasks can significantly affect final user satisfaction. In addition, the actual order of tasks in the UX can significantly impact the prediction of final user satisfaction.

*4.2. Main Experiment*

In our preliminary studies, we found that changing the sequential order of tasks can significantly affect the prediction of final user satisfaction. In our main experiment, we aimed to demonstrate a new approach that uses the sequential task order to predict final user satisfaction based on UX related to randomly ordered tasks. We wanted to determine whether accounting for the sequence of actions can improve the prediction of final user satisfaction.

Sixty university students aged 20–25 years were enrolled in this experiment. Before starting, we explained that the participant's task was to find a once-in-a-lifetime trip they wanted to go on using the menu links on a travel agency's webpage. This website was created as a virtual service for our experiment only. The experiment involved three main tasks (A: Finding a tour; B: Finding a hotel; C: Reviewing the information) as shown in Figure 10 and allowed participants to test the website by selecting from the available menus. Participants were not limited in the amount of time they had to complete the tasks. The participants took, on average, approximately 17 min to complete all tasks (A + B + C) attentively.

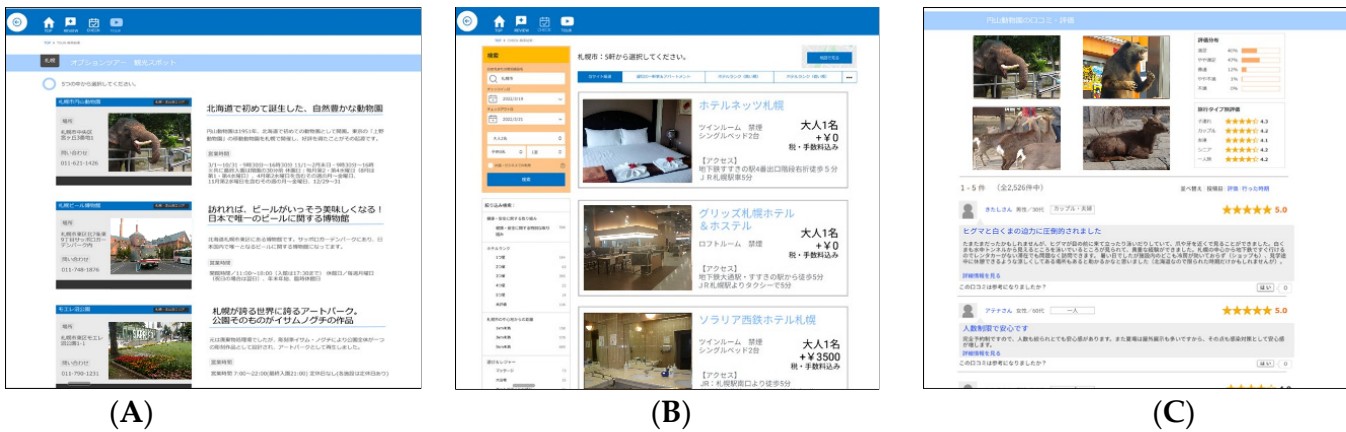

**(A)**　　　　　　　　　　　**(B)**　　　　　　　　　　　**(C)**

**Figure 10.** Examples of the interface of the travel agency website. (**A**) Task A: Tour finding; (**B**) task B: Hotel finding; (**C**) task C: Information review.

Each main task was further divided into three subtasks to obtain more specific data for building our machine learning models. The subtasks were a list of fix-ordered tasks. For example, Task B consisted of subtask B1, subtask B2, and subtask B3. As each main task comprised three subtasks, the total number of subtasks performed by each participant was nine. The participants had to record their satisfaction after completing each subtask (they had to evaluate nine subtasks). Once they had completed all tasks, they recorded their final user satisfaction, as shown in Table 3.

**Table 3.** Each main task was divided into three subtasks.

| Main Task A:<br>Finding a Tour | Main Task B:<br>Finding a Hotel | Main Task C:<br>Reviewing Information |
|---|---|---|
| Subtask A1: view tours | Subtask B1: view hotels | Subtask C1: read trip reviews |
| Subtask A2: read tour details | Subtask B2: read hotel details | Subtask C2: read tour reviews |
| Subtask A3: compare and book a tour | Subtask B3: compare and book a hotel | Subtask C3: read hotel reviews |

They were requested to record the order of their activities and their satisfaction during and after they had completed all tasks. Six groups comprising 10 participants each were assigned a different set of ordered main tasks to perform. A balanced distribution of participants in each group is expected to reduce inequivalence across variables when building the model.

As participants completed the three main tasks in random order, a customized UX Curve was progressively constructed to record when Tasks A, B, and C were performed and the participant's level of satisfaction at each point, as shown in Figure 11. This procedure is often implemented in visits to travel websites in participant experiments. After completing the three main tasks, participants recorded their final satisfaction based on their experiences. It should be noted that final user satisfaction after the three main tasks was evaluated for the study only and did not affect the UX Curve for the website itself. The obtained final user satisfaction data were used as a target class variable for supervised learning. For UXE, we used a 5-point scale that ranged from −2 to +2 (Figure 11, right) based on a previously reported UX graph template [14]. Thus, the number of classes used in this study was five.

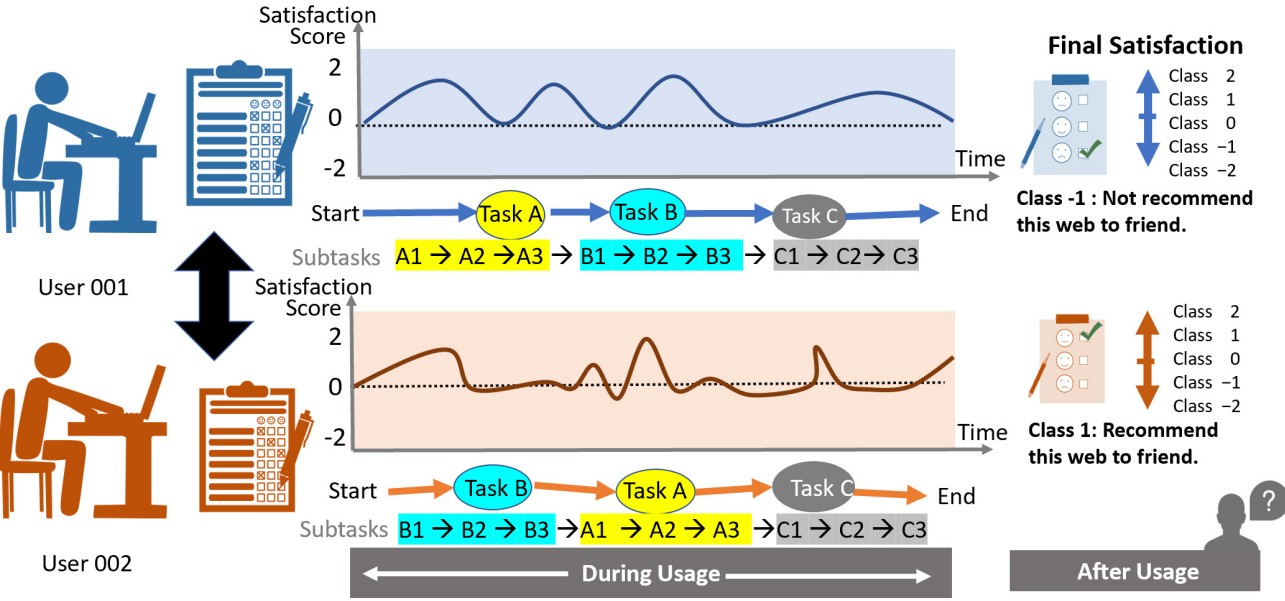

**Figure 11.** Example of a UX curve based on randomly ordered tasks.

### 4.2.1. Dataset Structure

An essential phase in our approach involved preparing the data for building machine learning models, in which the UX data were transformed into a feature matrix. As we wanted to study the usefulness of sequential ordering for identifying factors predictive of final user satisfaction, we created two different datasets. Dataset I contained UX data that did not account for the actual order of the tasks performed on the travel agency website, while Dataset II contained a randomly ordered UX. Moreover, Dataset II accounted for the actual order of data as it comprised satisfaction score data that were sorted by the actual order of tasks performed. For example, Figure 12 shows that, in Dataset II, the first user (User01) gave a satisfaction score of 0 after completing task B1. They then gave a satisfaction score of 1 after completing task B2, and then a satisfaction score of 0 after completing task B3. In our study, we compared the prediction results between Dataset I, which did not account for the actual task order, and Dataset II, which did.

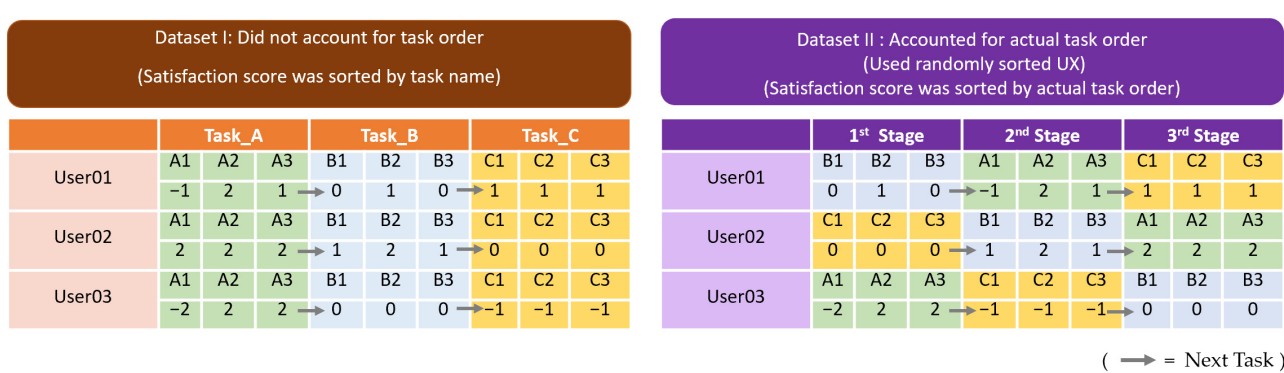

**Figure 12.** Two datasets created based on task order.

### 4.2.2. Building Classification Models

As mentioned above, several studies have used classification algorithms to predict final user satisfaction [1,16–19,22]. We tested four SVM algorithms [26], namely SVM linear, SVM sigmoid, SVM RBF, and SVM polynomial, and random forest, KNN, and AdaBoost. We created and trained these classification models using two datasets: A training set and a test set, before comparing their performance to identify the best model for predicting final user satisfaction.

Because the original dataset, which was based on the 5-point scale, produced an accuracy score of less than 0.50, we concluded that the results obtained using the 5-point scale were insufficiently accurate. Thus, we converted the 5-point scale ($-2$, $-1$, 0, +1, +2) to a 3-point scale ($-1$, 0, +1). To do this, we first merged the $-2$ and $-1$ point classes into a single "$-1$ point" class and grouped the +2 and +1 point classes into a "+1 point" class. The final 3-point scale comprised the target classes $-1$, 0, and +1 points. One advantage of rescaling the 5-point scale to a 3-point scale is the increased number of samples per class, which can be useful for building machine learning models.

In the main experiment, we noted an unequal distribution of classes within Dataset I and Dataset II. Generating data for the minority class, defined as that with the smallest sample size of all the classes, is a challenging problem for training in machine learning. One way to solve this problem is to oversample samples in the minority class. To do this, we used the SMOTEN [31] oversampling technique in the evaluation step. As indicated in Section 2.4, multiple techniques exist for dealing with imbalanced sample distributions. Oversampling the minority class is one such approach used in data science [31].

### 4.2.3. Model Evaluation

In the evaluation step, we measured each classification model's efficiency in terms of accuracy, precision, and recall. The conventional leave-one-out cross-validation method was used to evaluate performance [36], in which one set of data is left out of the training set. For example, of the original data from 10 users, those from 9 were used to train the model, and one was used for validation. The leave-one-out cross-validation (LOOCV) procedure is appropriate for small datasets because its results are reliable and unbiased in estimating model performance [37].

### 5. Results and Discussion

We conducted experiments to test our hypothesis that machine learning models that account for the sequential order of tasks performed when using a product or service produce more accurate predictions of final user satisfaction than those that do not. By comparing the predictive performance of Datasets I and II using seven machine learning algorithms, we showed that, indeed, sequential ordering was important for prediction accuracy. Among the models tested, we found that the machine learning classification model produced the greatest accuracy, as shown in Table 4.

**Table 4.** Performance of test models with SMOTEN oversampling.

| | Scores | Dataset | Random Forest | KNN | SVM Poly | SVM Linear | SVM RBF | SVM Sigmoid | AdaBoost |
|---|---|---|---|---|---|---|---|---|---|
| LOOCV | Cross-Validation Accuracy | Dataset I | 0.68 | 0.61 | 0.68 | 0.60 | 0.75 | 0.46 | 0.61 |
| | | Dataset II | 0.70 | 0.71 | 0.76 | 0.76 | 0.76 | 0.61 | 0.70 |
| Split for training/test (80/20) | Accuracy | Dataset I | 0.83 | 0.90 | 0.93 | 0.83 | 0.93 | 0.57 | 0.87 |
| | | Dataset II | 0.83 | 0.83 | 0.97 | 0.83 | 0.93 | 0.70 | 0.73 |
| | Precision | Dataset I | 0.85 | 0.92 | 0.93 | 0.87 | 0.94 | 0.54 | 0.89 |
| | | Dataset II | 0.88 | 0.85 | 0.97 | 0.88 | 0.94 | 0.84 | 0.80 |
| | Recall | Dataset I | 0.83 | 0.90 | 0.93 | 0.83 | 0.93 | 0.57 | 0.87 |
| | | Dataset II | 0.85 | 0.83 | 0.97 | 0.83 | 0.93 | 0.70 | 0.73 |

SVM = support vector machine; Poly = polynomial kernel; KNN = K-nearest neighbors, RBF = radial basis function; Dataset I = Did not account for task order; Dataset II = Accounted the actual task order; Accuracy score values were between 0.00 to 1.00, with a higher value indicating higher accuracy.

### 5.1. Accounting for Actual Task Order in Randomly Ordered UX

In LOOCV, the highest leave-one-out cross-validation accuracy was obtained for Dataset II at 76%, which was significantly higher than that for Dataset I at 68%. This difference is thought to be partly due to the difference in the structure of the datasets in terms of the task order considered in the data, as shown in Figure 12. Moreover, we compared the prediction results obtained using the split validation technique between

the two datasets and found that Dataset II demonstrated significantly better performance, producing the highest accuracy of 97% compared to 93% for Dataset I. Thus, our initial findings showed that evaluation based on Dataset II, which accounted for the actual task order, may be better for predicting satisfaction levels when estimating final user satisfaction. Therefore, this study confirmed that task order in UXE may directly affect final user satisfaction.

*5.2. Machine Learning Algorithms*

The best machine learning algorithm with which to predict final user satisfaction for the travel agency website was polynomial kernel SVM, which had an accuracy of 97%. Although the other machine learning algorithms tested, when used with Dataset II, all generated prediction accuracies of more than 80%, we propose that the polynomial kernel SVM may be most accurate for the prediction of satisfaction level based on randomly ordered tasks because the higher-degree polynomial kernel in the SVM algorithm allows for a more flexible decision boundary [38]. In this case, the polynomial kernel had three parameters (offset, scaling, degree), which are relatively easy to fine tune to obtain classification results with the highest accuracy.

## 6. Conclusions

This paper presents a new approach using machine learning techniques to predict final user satisfaction based on UX related to randomly ordered tasks. The main experiment confirmed the effectiveness of accounting for task order when predicting user satisfaction as an indicator of UX. The use of Dataset II, which accounted for the actual order of the tasks performed by users, in the measurement of satisfaction provided the highest cross-validation accuracy of 76% when compared to Dataset I, which did not account for the task order. Further, we showed that the polynomial kernel SVM produced the most accurate predictions among the machine learning methods tested.

The main finding of our study was that accounting for the actual order or sequence of actions can improve predictions of final user satisfaction. Given that the UX in real life involves randomly ordered tasks, our proposed method reflects the real-world setting. We expect that our findings will help other researchers conducting UXE to obtain more accurate predictions of user satisfaction. Our study contributes to knowledge in the field in various ways. First, our contribution relates to the outcomes of UX. Data on randomly ordered tasks or random ordering are often difficult to understand and analyze. We identified a relationship between the order or sequence of actions and episodic or cumulative UX, which is related to final user satisfaction. Our study shows that understanding the order or sequence of actions as the UX curve changes [12,20,21] can help determine final user satisfaction. Second, machine learning algorithms such as SVM can be used to accurately predict final user satisfaction and contribute to developing better products through the analysis of randomly ordered UX. Hence, it is important to carefully monitor randomly ordered UX. Third, accounting for the actual order or sequence of actions performed by users of a product or service could constitute a new approach for improving the predictive accuracy of final user satisfaction.

Future research should confirm these initial findings by using random ordering in the UX to assess underlying factors for predicting final user satisfaction with other products. The small number of participants and the small number of tasks in the main experiment are limitations of this study. Further validation of the methods is needed using larger sample sizes and a greater number of randomly ordered tasks. In future work, it may also be possible to introduce other measures and features such as operation time data on each task to improve the predictive performance for final user satisfaction reported here.

**Author Contributions:** Conceptualization, K.K. and N.N.; methodology, K.K., D.H., V.Y. and N.N.; software, K.K.; validation, K.K.; formal analysis, K.K.; investigation, K.K.; resources, K.K. and N.N.; data curation, K.K.; writing—original draft preparation, K.K.; writing—review and editing, K.K. and N.N. All authors have read and agreed to the published version of the manuscript.



**Funding:** This research was funded by JSPS KAKENHI, grant number JP20K12511 and local-5G project of Tokyo Metropolitan University.

**Institutional Review Board Statement:** The study was approved by the Ethics Committee of Faculty of Systems Design, Tokyo Metropolitan University (Protocol code H21-016 on 2021/8/2 and Protocol code H22-009 on 27 April 2022).

**Informed Consent Statement:** Informed consent was obtained from all subjects involved in the study.

**Data Availability Statement:** Not applicable.

**Acknowledgments:** The authors also thank Misaki Imamura and Ayato Kakegawa for supporting experiments.

**Conflicts of Interest:** The authors declare no conflict of interest.

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
