# Peer review of "Using Random Ordering in User Experience Testing to Predict Final User Satisfaction"

_informatics, doi:10.3390/informatics9040085_

Round 1
Reviewer 1 Report
The authors present a method to predict final user satisfaction based on random ordering in User Experience testing. The study examined seven machine learning algorithms for classification of UX data, namely random forest, K-nearest neighbors, SVM polynomial, SVM linear, SVM radial basis function (RBF), SVM Sigmoid and AdaBoost. I have read this manuscript and I quite interest as a manuscript describes well enough. I can understand the novelty of this study well.Challenging problem of this study was explained clearly in the 2.2) Order Effect in UX section. The idea of accounting for actual task order is interesting and may be efficient for the final satisfaction prediction.
In literature review, there are many evidences and the related works is described. The authors can summarize the related works well.
In experiment, the authors tested their idea on preliminary experiment I and II before starting a main experiment. Therefore, this study results could give us a good evidence that confirmed the actual order or sequence of actions in UXE could directly affect the prediction of final user satisfaction. However, I have minor comments that authors should improvement. Main: - Minor: (1) At Line 138, please improve the quality ofFigure 3. (2) At Line 145, please cite of Sukamto et al. research at the end of sentence. (3) At Line 292 (Figure 6) , proposed workflow is easy to understand. However, the author should add algorithm name: Random forest, K-nearest neighbors, AdaBoost in "Build Classification Model" of Figure 6. (4) At Line 412, in main experiment, did author used a limited time in User Experience testing? How about is total usage time of all tasks (A+B+C)? (5) (Figure 11) I recommend author should add axis title, please add x-axis title and y-axis title of UX Curve (6) At Line 466, There are wavy red underlines on left of figure. 12, please improve it. (7) At Line 511, please use decimal number ("0.00 to 1.00") (8) At Line 544, the main finding of this study was that accounting for the actual order or sequence of actions can improve predictions of final user satisfaction. It is good if authors mention that this finding will be useful to the related researchers in User Experience testing. or please add practical implication. (9) At Line 557, in future work, it is better if authors increase the number of participants and number of tasks in User Experience testing.
Reviewer 2 Report
The conclusion part can be improved.
Reviewer 3 Report
This paper applies SVM to user satisfaction. It lacks substantial novelty in the following points.
First, it seems that user satisfaction problem can be easily resolved without using machine learning technique. Authors should clarify why machine learning is imperative to address user satisfaction problem and the main advantage in adaptation of machine learning to user satisfaction problem when compared to other techniques.
Second, the proposed scheme is not compared to the related work. If the comparison is not possible, authors should justify the main reason for non-comparison.
Reviewer 4 Report
The article discusses a new approach to predicting the end user satisfaction based on UX associated with randomly ordered tasks. In the introduction, the authors review existing work on both end-user satisfaction analysis in general and tools to improve the accuracy and quality of the assessment. The content of the work allows to conclude that the authors are highly qualified, as well as the high quality of the presented results. The conducted experiments are described in detail that makes it possible to conclude that the experiments are reliable. The text is accompanied by a large number of diagrams, charts and graphs, which increases the visibility of the material. I think that the work deserves publication.
